# A Comprehensive Review on the Benefits and Problems of Curcumin with Respect to Human Health

**DOI:** 10.3390/molecules27144400

**Published:** 2022-07-08

**Authors:** Siyu Liu, Jie Liu, Lan He, Liu Liu, Bo Cheng, Fangliang Zhou, Deliang Cao, Yingchun He

**Affiliations:** 1Post-Graduate School, Hunan University of Chinese Medicine, Changsha 410208, China; liusiyu20212013@163.com (S.L.); liujie20202015@163.com (J.L.); liuliu20222014@163.com (L.L.); zhoufangliang@hncm.edu.cn (F.Z.); 2The First Clinical College of Traditional Chinese Medicine, Hunan University of Chinese Medicine, Changsha 410007, China; island365@163.com (L.H.); chengbo13974980387@163.com (B.C.); 3Hunan Provincial Engineering and Technological Research Center for Prevention and Treatment of Ophthalmology and Otolaryngology Diseases with Chinese Medicine and Protecting Visual Function, Hunan University of Chinese Medicine, Changsha 410208, China; 4Hunan Provincial Key Laboratory for the Prevention and Treatment of Ophthalmology and Otolaryngology Diseases with Traditional Chinese Medicine, Hunan University of Chinese Medicine, Changsha 410208, China

**Keywords:** curcumin, malignant tumors, Alzheimer’s disease, blood diseases, viral infectious diseases, hurdles, potentials and side effects

## Abstract

Curcumin is the most important active component in turmeric extracts. Curcumin, a natural monomer from plants has received a considerable attention as a dietary supplement, exhibiting evident activity in a wide range of human pathological conditions. In general, curcumin is beneficial to human health, demonstrating pharmacological activities of anti-inflammation and antioxidation, as well as antitumor and immune regulation activities. Curcumin also presents therapeutic potential in neurodegenerative, cardiovascular and cerebrovascular diseases. In this review article, we summarize the advancements made in recent years with respect to curcumin as a biologically active agent in malignant tumors, Alzheimer’s disease (AD), hematological diseases and viral infectious diseases. We also focus on problems associated with curcumin from basic research to clinical translation, such as its low solubility, leading to poor bioavailability, as well as the controversy surrounding the association between curcumin purity and effect. Through a review and summary of the clinical research on curcumin and case reports of adverse effects, we found that the clinical transformation of curcumin is not successful, and excessive intake of curcumin may have adverse effects on the kidneys, heart, liver, blood and immune system, which leads us to warn that curcumin has a long way to go from basic research to application transformation.

## 1. Introduction

Turmeric has a long history. Early in the 12th century, Marco Polo mentioned the medicinal value of turmeric in the Sino-Indian voyage diary, and in the 13th century, turmeric was introduced from India into Europe by Arab merchants. During British rule in India in the 15th century, curry powder mixed with turmeric and several other herbs was first used for medical purposes in humans [1]. In traditional Chinese medicine, turmeric is used to promote blood circulation and release blood stasis. Turmeric grows in hot and humid climates and therefore widely exists in tropical and subtropical regions, especially in India, China and Southeast Asia. Modern botany classifies turmeric, zedoary and yujin into the turmeric family.

Turmeric is mainly composed of curcumin compounds (including curcumin, demethoxycurcumin and bisdemethoxycurcumin), volatile oil and resins [2]. Among them, curcumin separated from the dried rhizomes of turmeric is the main active component of turmeric, with a molecular formula of C_21_H_20_O_6_ (Figure 1). Curcumin is a tasteless, orange-red photosensitive powder that is insoluble in water. To date, curcumin has shown multiple pharmacological activities, such as anti-inflammation [3], antioxidation [4], antitumor [5] and immune regulation activities [6]. Therefore, curcumin is considered to have a wide range of therapeutic potentials in solid tumors, mental cognitive disorders, and cardiovascular and cerebrovascular diseases. Databases such as Web of Science, PubMed, Google Scholar and CNKI were used to perform a literature search, using terms that define curcumin and disease. The literature search covered studies including basic research and reviews published from 2000 to 2022. We found that a large number of literature reports were focused on malignant tumors, AD and blood abnormalities, with a small part of the curcumin antiviral ability elaborated. With a focus on advances with respect to curcumin for antitumor, AD, viral infection and blood disease applications, in this paper, we review its pharmacological effects and mechanistic approaches. In addition to reviewing the outstanding potential of curcumin in a variety of diseases, we also focuses on the problems associated with curcumin from basic research to clinical translation, such as its low solubility, leading to poor bioavailability, as well as the difference in the purity of curcumin in the edible and medicinal fields, which may affect its maximum effect. A lack of high-quality clinical research led the advancement of studies on curcumin stagnate atthe preclinical stage, so its adverse effects still need to be explored. Through a literature review, we identified case reports showing that excessive curcumin intake may have adverse effects on the kidneys, heart, liver, blood and immune system, which serves as a warning that curcumin still has a long way to go from basic research to application and transformation.

## 2. Curcumin in Treatment of Malignant Tumors

Tumors are organisms formed by the long-term synergistic action of various tumorigenic factors inside and outside the body, in which local tissue cells lose their normal control over their growth at the gene level, resulting in abnormal cell proliferation. Cancer has gradually replaced cardiovascular disease as the leading cause of death in developed countries. Although breakthrough progress has been made in targeted and immune therapies against cancer, drug resistance and side effects are the main drawbacks, and cancer mortality is still high. Development of therapeutic agents from natural plants remains one of the major efforts in the field cancer therapy. To date, curcumin has exhibited a wide range of antitumor activity, inhibiting proliferation, invasion, metastasis and angiogenesis, as well as inducing apoptosis and autophagy, making the body sensitive to radiotherapy and chemotherapy.2.1. Inhibition of Cell Proliferation and Induction of Apoptosis

Imbalance between infinite cell proliferation and apoptosis is the main cause of malignant tumors, which is regulated by such representative genes as the Bcl-2/Bax complex, the caspase family, p53, p21, survivin, cyclin D1, proliferative cell nuclear antigen (PCNA), etc. Curcumin may inhibit the proliferation and induce the apoptosis of colon cancer cells by promoting Bax and caspase-3 expression but inhibiting the expression of Bcl-2, cyclin D1 and survivin [7,8]. Through similar mechanisms, curcumin may exert anticancer activity in thyroid cancer [9], lung cancer [10] and renal cancer [11].

In recent years, a number of studies have focused on the functions of microRNAs (miRNAs) in tumorigenesis. miRNAs are single-stranded, non-coding, endogenous molecules containing 18–25 nucleotides widely participating in the regulation of gene expression. Approximately 50% of annotated miRNAs are located in fragile sites related to tumors in the genome. Curcumin can affect the expression of both cancer-suppressive and cancer-promoting miRNAs in malignant tumors, thus controlling cell proliferation and apoptosis. For instance, in prostate cancer cells, curcumin inhibits cell proliferation through miR-145-mediated downregulation of cyclin D1 and cyclin-dependent kinases 4 (CDk4) or miR-30a-5p-mediated suppression of PCNA clamp-associated factor (PCALF) [12]. Curcumin may also upregulate miR-34a to inhibit β-catenin signaling activity and therefore the expression of c-Myc, cyclin D1 and PCNA [13]. Curcumin may also inhibit prostate cancer cell proliferation through miR-143-mediated mechanisms to downregulate phosphoglycerate kinase 1(PGK1) [14]. Through modulation of the miR-21/tissue inhibitors of metalloproteinase 3 (TIMP3) and miR-21-5p/SOX6, curcumin inhibits the proliferation, migration and invasion of hepatocellular carcinoma cells [15]. Curcumin can also affect the proliferation and apoptosis of colon cancer cells by inhibiting the Wnt/β-catenin signaling pathway by downregulating miR-130a [16] or by regulating the miR-491/PEG10 pathway [17]. Curcumin also inhibits the proliferation of colon cancer cells through a miR-199b-5p-mediated mechanism to downregulate p21-activated kinase 4 (PAK4), phospho-mitogen-activated protein kinase (p-MEK), MEK, phospho-extracellularly regulated protein kinases (p-ERK) and ERK [18]. Curcumin also upregulates miR-192-5p to inhibit the activity of the Wnt/β-catenin signaling pathway and the proliferation of non-small cell lung cancer [19], in addition to inducing apoptosis of cancer cells. For instance, curcumin can induce apoptosis of bladder cancer cells by downregulating miR-7641 [20] or miR-1246 [21]. Curcumin can inhibit the TLR4/NF-κB signaling pathway by downregulating miR-210 and thus inducing apoptosis of prostate cancer cells [22]. In gastric cancer cells, curcumin can upregulate phosphate and tension homology deleted on chromosome ten (PTEN) through a miR-21-mediated mechanism to inhibit Akt activity and induce apoptosis [23]. Curcumin can also downregulate miR-133a-3p [24] or upregulate miR-33-b to inhibit the expression of X-linked inhibitor of apoptosis protein (XIAP), inducing gastric cancer cell apoptosis [25]. Curcumin also upregulates miR-9 to promote the expression of caspase-3 and induce apoptosis of ovarian cancer cells [26]. In addition, curcumin can affect angiogenesis by upregulating miR-29 and downregulating vascular endothelial growth factor (VEGF) in hepatocellular carcinoma [27].

In addition to miRNAs, curcumin may also inhibit the proliferation of malignant cells and induce apoptosis by regulating long, non-coding RNAs (lncRNAs). Guo et al. [28] used curcumin to stimulate liver cancer cells and found expression changes of 128 lncRNAs, of which the upregulation of lncRNA AK125910 was most impressive. Subsequent in vitro studies confirmed that curcumin can upregulate lncRNA AK125910 to induce apoptosis. In addition, curcumin may downregulate lncRNA UCA1 to inhibit lung cancer cell proliferation and induce apoptosis [29]. Curcumin also upregulates lncRNA NBR2, inhibiting the Akt/mTOR signaling pathway and inducing apoptosis [30]. Figure 2 partly presents representative miRNAs and lncRNAs that are regulated by curcumin, through which curcumin inhibits cancer cell proliferation or induces apoptosis.

### 2.1. Induction of Autophagy

Autophagy, also known as type II programmed cell death, is a process by which cells degrade their own damaged organelles or macromolecules by lysosomes under the regulation of autophagy-related gene Beclin-1. Autophagy is closely related to apoptosis. Moderate autophagy can protect cells, whereas excessive digestion and degradation of organelles cause the death of autophagic cells, i.e., autophagy-induced apoptosis [31]. Antiapoptotic protein Bcl-2 inhibits autophagy and apoptosis by isolating Beclin-1 under nutrient-rich conditions.

Kim et al. [32] used curcumin to treat oral squamous cell carcinoma (OSCC) cells. Acridine orange and MDC staining was used to detect autophagic vacuoles, and Western blotting was applied to evaluate the conversion of LC3-I to LC3-II. Results showed the formation of active autophagosomes, and curcumin-induced apoptosis was inhibited by autophagy inhibitors. The autophagy induced by curcumin in malignant tumor cells may mainly rely on the following three pathways: ① Inhibition of the PI3K/Akt/mTOR signaling pathway: Curcumin may induce autophagy in colon cancer cells HCT116 and SW620 by downregulating the activity of the Yes-associated protein (YAP) through inhibition of the IGF1/PI3K/Akt/mTOR pathway [33]. In prostate cancer cells LNCaP, curcumin upregulates Beclin-1 and LC3-II, inducing autophagy through inhibition of the Akt/mTOR pathway [34], and in ovarian cancer cells SK-OV-3 and A2780, curcumin induces protective autophagy through inhibition of the AKT/mTOR/p70S6K pathway [35]. In addition, curcumin induces autophagy in human melanoma cells A375 and C8161 by upregulation of Beclin-1 [36] and in gastric cancer cells SGC-7901 and BGC-823 via upregulation of p53 and p21 through inhibition of the PI3K/Akt/mTOR pathway [37]. ② Activation of the MAPK/ERK signaling pathway: Curcumin activates this pathway through induction of ROS and triggers autophagy in human colon cancer cells HCT116 [38]. Chen et al. [39] also reported that curcumin might induce autophagy of lung adenocarcinoma cells A549 by activating the MAPK/ERK signaling pathway. ③ Activation of the AMPK signaling pathway: In human lung adenocarcinoma cells A549, AMPK kinase was activated by curcumin to induce autophagy [40]. Figure 2 partly summarizes the mechanisms by which curcumin induces autophagy in cancer cells.

### 2.2. Inhibition of Invasion and Metastasis

Invasion and metastasis are characteristic of malignant tumor progression and poor prognosis. Curcumin inhibits the invasion and metastasis of malignant cells in a wide range of cancers, such as lung cancer [41,42], colon cancer [43,44] and cervical cancer [45] by reversing epithelial–mesenchymal transition (EMT) or downregulating the expression of matrix metalloproteinases (MMPs). EMT is among the important means by which epithelial tumor cells acquire invasive and metastatic abilities. The loss of epithelial marker protein E-cadherin weakens the adhesion of epithelial, causing them to easily detach. Upregulation of interstitial marker proteins N-cadherin and vimentin causes the cells to acquire interstitial phenotypes and move easily [46]. MMPs degrade the extracellular matrix, which is beneficial to invasion and metastasis of malignant tumor cells [47].

Recent studies show that curcumin may affect the EMT process and MMPs secretion through miRNA- and lncRNA-mediated mechanisms. For example, curcumin may inhibit the expression of JAK/STAT signaling effector proteins p-JAK1, JAK1, p-JAK2, JAK2, p-JAK3, JAK3, p-STAT1, STAT1, p-STAT2 and STAT2 by upregulating miR-301a-3p, thus affecting the vitality of thyroid cancer cells TPC-1 [48]. Cai et al. [49] found that lncRNA H19 can activate the EMT process, as well as invasion and metastasis of breast cancer MCF 7/TAMR cells, whereas curcumin can reverse these phenotypes induced by lncRNA H19. Yin et al. [50] reported that curcumin may interfere with the interaction between Gli1 in the Shh signaling pathway and β-catenin in the Wnt signaling pathway, upregulating the EMT markers, thus inhibiting the invasion and metastasis of gastric cancer cells SGC-7901.

The tumor microenvironment consists of stromal cells and factors that are stable in the stromal environment and is considered the soil for tumor growth [51]. EMT-inducing factors, such as TGF-β, PI3K/Akt and Wnt, activated by the tumor microenvironment can enhance the invasion ability of tumor cells. A single-layer tumor microenvironment coculture model of colon cancer HCT-116 cells and MRC-5 fibroblasts was proposed, and it was found that TGF-β3, metastatic active adhesion molecules, proliferation-related proteins and EMT-related factors in HCT-116 cells were significantly upregulated. The high-density tumor microenvironment coculture model showed that tumor-promoting factors, TGF-β3 and EMT-related factors were also upregulated; that is, there might be crosstalk between tumor cells and the tumor microenvironment, and curcumin can significantly reduce this crosstalk [52].

In addition to EMT and MMP secretion, which are recognized as important processes of invasion and metastasis of malignant tumor cells, the high expression of the actin-binding protein fascin is also closely related to the metastasis and recurrence of malignant tumors, leading to high mortality. Fascin affects microfilaments by producing filamentous feet and platy feet, playing the role in invading cells. Kim et al. [53] found that the expression of STAT3 and fascin protein in ovarian cancer cells SKOV3 treated with curcumin was downregulated so that the adhesion ability, scratch-healing ability and membrane-penetrating ability all decreased. Immunofluorescence showed that the cell morphology changed to a polygonal shape, and the formation of filopodia significantly decreased. In addition, curcumin can inhibit invasion and metastasis of ovarian cancer cells by inhibiting phosphorylation of focal adhesion kinase (FAK) [54].

### 2.3. Inhibition of Tumor Angiogenesis

Vascular endothelial cells degrade the extracellular matrix and form new blood vessels in malignant tumors, providing rich nutrients for tumor cells, as well as channels for spreading, thus promoting invasion and metastasis. Both in vivo and in vitro experiments prove that curcumin can inhibit tumor angiogenesis by downregulating the expression of vascular-related growth factors, such as VEGF, Ang-1, Ang-2, PDGF, COX-2, HIF-1α, TGF-β and bFGF [55]. Recent studies suggest that mimicry of angiogenesis and mosaic-like vascular patterns are among the main sources blood supplies for solid tumors. Some malignant tumor cells may function as endothelial cells and form new microcirculation lumens with the extracellular matrix, which accelerates the development of tumors. The vascular mimicry may interact with endothelial cells to form reticulated blood vessels, namely a mosaic phenomenon. In the model of vascular mimicry and mosaic of human sarcoma cells S180, the number of tubular structures and the expression of MMP-2 and MMP-9 proteins were significantly decreased by curcumin [55]. In vivo experiments also showed that curcumin can inhibit the growth of subcutaneously transplanted sarcoma in mice and the formation of tumor blood vessels with a decrease in the density of microvessels [56]. After high-dose curcumin intervention, CD31 expression in malignant pleural mesothelioma tumors decreased significantly, suggesting that curcumin inhibited angiogenesis in tumors [57] (Figure 2). In addition, Fan’s group [58] found that curcumin may directly or indirectly inhibit neutrophil elastase (NE) and bidirectionally regulate angiogenesis through the HIF-1α/mTOR/VEGF/VEGFR pathway. Therefore, in an animal model of transplanted lung cancer complicated with ischemia, curcumin not only promotes blood flow reconstruction of ischemic tissues but also inhibits the angiogenesis of lung cancer, which suggests the potential value of curcumin in the treatment of malignant tumors complicated with ischemia.

### 2.4. Sensitization of Cancer Treatment

During the treatment process of malignant tumors, their sensitivity to chemotherapy and radiotherapy often decreases with time, and the application of natural chemical components to reverse drug resistance is an important strategy for resensitization [59]. Curcumin can dose-dependently inhibit the proliferation of cisplatin-resistant colon cancer cells HCT8/DDP and promote apoptosis by inhibiting the expression of lncRNA KCNQ1OT1, whereas the targeted expression of KCNQ1OT1 prevents the effect of curcumin on HCT8/DDP cells by promoting the binding of miRNA-497 to Bcl-2, thus eliminating the inhibitory effect of curcumin on tumorigenesis in vivo [60]. Liu et al. [61] constructed paclitaxel-resistant human ovarian cancer cells OVCAR3/T with strong autophagy and found that curcumin led to a dose-dependent reduction in autophagy of the drug-resistant cells and that combination with paclitaxel significantly reduced cell viability. Huang et al. [62] used curcumin and cisplatin alone or in combination to treat multi-drug-resistant gastric cancer cells BGC-823/5-Fu and found that the number of cell colonies and transmembrane cells in the combination intervention was lower than that.only with cisplatin In the cells treated with curcumin plus cisplatin, the expression of E-cadherin protein was upregulated, and the expressions of Vimentin, Wnt2 and β-catenin proteins was downregulated, suggesting that curcumin might inhibit the invasion and metastasis of drug-resistant gastric cancer cells by downregulating the Wnt/β-catenin signaling pathway, thereby enhancing the sensitivity of gastric cancer cells to cisplatin.

5-Fu is another of the most widely used chemotherapeutic drugs in malignant tumors, and it is converted into 5- fluorouracil deoxynucleotide (5F-dUMP) in cells to inhibit deoxythymidylate synthase, thereby affecting the synthesis of DNA. Karthika et al. used curcumin combined with 5-Fu (1:4), resulting in an improved antitumor effect on a rat colon cancer model induced by titanium dioxide nanoparticles (TiO_2_NPs) and dimethylhydrazine (DMH) compared to 5-Fu alone [63]. Mehdi Shakibae et al. cocultured HCT-116 and HCT-116R using a 3D model. The results showed that compared with 5-Fu alone, curcumin combined with 5-Fu reduced the IC_50_ of 5-Fu to HCT-116 from 8 nM to 0.8 nM (HCT-116) and 0.1 nM (HCT-116R), respectively, confirming the enhanced sensitivity of colon cancer cells to 5-Fu [64]. Shakibaei M et al. [65] used a similar high-density 3D model to culture colon cancer cells HCT-116, MMR-deficient HCT-116 and their corresponding drug-resistant cells (HCT-116R, HCT-116 + ch3R) and found that curcumin combined with 5-Fu significantly enhanced colony division, induced cell apoptosis and inhibited cell proliferation. Furthermore, the sensitivity of 5-Fu was strengthened, which provided a far-reaching influence on the chemotherapy resistance of MMR-deficient colon cancer. The combination of 10μM curcumin and 10 μM 5-Fu may resensitize breast cancer cells to 5-Fu by downregulating thymidylate synthase-dependent nuclear factor-κB (NF-κB) [66]. The latest research showed that the medium collected from cancer-associated fibroblasts (CAF) can increase the resistance of gastric cancer cells to 5-Fu by activating the JAK/STAT3 signaling pathway, and curcumin may reverse the 5-Fu resistance induced by CAF by inhibiting this signaling pathway [67].

In addition, curcumin can reverse the resistance of colon cancer cells to oxaliplatin [68], irinotecan [69], 5-Fu [70], doxorubicin [71], capecitabine and taxol [72], liver cancer to sorafenib [73], gastric cancer to trastuzumab [74] and 5-Fu [75], esophageal cancer to vincristine [76], pancreatic cancer to gemcitabine [77], breast cancer [78] and thyroid cancer [79] to doxorubicin, and lung cancer to paclitaxel [80]. Curcumin inhibits the proliferation, invasion and metastasis of these drug-resistant cell lines and promotes their apoptosis through a wide range of targets. Figure 2 partly demonstrates the effective mechanisms by which curcumin resensitizes drug-resistant cancer cells.

In radiation treatment of malignant tumors, curcumin can improve the lethality of radiation against a variety of malignant tumor cells, thus reducing the adverse reactions of high-dose radiation. This may be related to curcumin-induced oxygen contents in tumor tissues, interfering with the DNA repair and cell cycle of damaged cells [81] and inducing tumor cell apoptosis [82]. Figure 3 demonstrates the effective mechanisms by which curcumin resensitizes radio-resistant cancer cells.

## 3. Curcumin in Treatment of AD

In the United States, AD is the third most expensive disease after malignant tumors and cardiovascular diseases [83]. Interestingly, studies have shown that the rate of memory decline in patients with malignant tumors before and after diagnosis is lower than that of healthy people. A diagnosis of malignancy is associated with an 11% reduction in the incidence of AD [84]. It is possible that the inhibition of peptidyl-prolyl cis-trans isomerase NIMA-interacting 1 (PIN1), which is responsible for the structural regulation of protein, hyperphosphorylated TAU and enhanced the risk of AD. The polymorphism associated with the reduced expression of PIN1 has been associated with a reduced risk of cancers [85]. Malignant tumors and AD may be natural enemies. However, with global aging, AD will one day represent a more considerable challenge to all mankind than malignant tumors.

Briefly, AD is a disease that seriously affects the physical and mental health of the elderly, with recent memory impairment and cognitive decline as the main symptoms [86]. Senile plaques formed by deposition of β-amyloid protein in the brain gather outside the nerve cells, nerve fibers formed by phosphorylated TAU protein gather inside the nerve cells and synaptic neurons are lost, all of which are typical pathological features of Alzheimer’s disease. It has been reported that curcumin can penetrate the blood–brain barrier and improve the memory and cognitive function of AD in models of mice, rats, cats and non-primates [87]. Curcumin functions mainly through the following mechanisms.

### 3.1. Reduction in Beta-Amyloid Deposition

Normally, β-amyloid is secreted from cells and degraded. In AD, secreted β-amyloid is aggregated into neurotoxic, insoluble plaques [88]. The keto or enol ring of curcumin interacts with the aromatic ring of β-amyloid protein, damaging the β-sheet, thereby inhibiting the generation and aggregation of senile plaques and promoting plaque decomposition [89,90]. Sun et al. [91] found that in curcumin-intervened mice, the Morris water maze detected improvement in memory. Immunohistochemical staining and RT-PCR showed that the positive intensity and mRNA levels of β-amyloid in the brain tissues of the mice were significantly reduced. Huang et al. [92] reported that curcumin might inhibit the NF-κB signaling pathway by selectively activating the estrogen receptor β (ERβ) and regulating the expression of β-site amyloid precursor protein-cleaving enzyme 1 (BACE1) at the promoter level, thereby inhibiting the amyloid cleavage pathway and reducing extracellular β-amyloid contents. Zhang et al. [93] proposed that curcumin could also reduce amyloid deposition by inhibiting the activity of glycogen synthase kinase 3 (GSK-3).

### 3.2. Prevention of TAU Protein Deposition

Tau protein is a neuron-predominant, microtubule-associated protein acting in proper function of the cytoskeletal network [94]. In AD patients, Tau protein is abnormally aggregated to form neurofibrillary tangles (NFTs) [95]. Curcumin can directly bind to the R2 site of the MTBR domain in the microtubule binding region of TAU, inducing conformational changes and increasing solubility, thus inhibiting the deposition of TAU protein in cells [96]. Curcumin may also decompose the formed TAU oligomers through regulation of the Nrf2/ARE pathway to reduce the oxidative stress damage of cells [97].

### 3.3. Regeneration of Synapses

Synapses are an important structure for the connection between synaptic neurons. Synapses are located in presynaptic vesicles and participate in the fusion of synaptic vesicles and presynaptic membranes, as well as the release of neurotransmitters, whereas shank and post-synaptic density (PSD) proteins located after synapses are involved in post-synaptic signal transduction and integration. Biopsies of the temporal cortex in patients with early AD showed that the density of synapses and neurons decreased. Xia [98], Wei [99] and others treated APP/PS1 transgenic mice with curcumin and found that the expression of synaptophysin, shank-1 and PSD-95 in the hippocampal CA1 area of mice increased. A labyrinth experiment showed that the learning and memory functions of mice improved.

Curcumin, as a strong antioxidant, may also prevent DNA damage and death of nerve cells by reducing intracellular oxidative stress and calcium ion flow [100,101]. Curcumin may also upregulate the expression of antioxidant proteins, maintain the structural integrity of mitochondria and strengthen the antioxidant capacity of mitochondria, thus preventing the damage of β-amyloid protein and other factors in cells.

## 4. Curcumin in Treatment of Abnormal Blood Conditions

### 4.1. Reduction in Blood Lipids

Abnormal blood lipids, especially the increase in low-density lipoprotein (LDL), are recognized as pathological factors of coronary atherosclerosis. By reducing blood lipids, the formation or rupture of plaques can be inhibited, and cardiovascular accidents are reduced. Meta-analysis showed that curcumin can significantly reduce serum low-density lipoprotein (LDL) and triglyceride levels in patients with hyperlipidemia but has no significant effects on serum HDL levels [102]. Through control of the absorption, accumulation and transport of cholesterol, the level of blood lipids can be effectively reduced. Zou et al. [103] fed ApoE −/− mice a high-fat diet alone or in combination with curcumin and found that cholesterol accumulation in the aorta, cholesterol absorption in the small intestine and the degree of atherosclerosis decreased by 56%, 51% and 45%, respectively. Kim et al. [104] found that curcumin can increase the expression of CYP7A1, a rate-limiting enzyme for cholesterol synthesis of bile acids, accelerating the excretion of cholesterol and thus reducing LDL, cholesterol and arteriosclerosis index in the liver of rats fed a high-fat diet.

Curcumin may also play an indirect role in the control of blood lipids through anti-inflammatory and antioxidant pathways. Fan et al. [105] intervened with curcumin in rabbit coronary atherosclerosis models and found that the blood lipid levels of rabbits decreased significantly, which was equivalent to the effect of rosuvastatin. In these models, the inflammatory indicators hs-CRP, TNF-α and IL-6 decreased, and SOD activity increased.

### 4.2. Reduction in Blood Sugar

Curcumin has limited effects on blood sugar of healthy people but can significantly reduce fasting blood sugar and postprandial blood sugar levels of patients with type II diabetes mellitus (T2DM) and significantly improve glycosylated hemoglobin levels [106]. Dysfunction of islet cells and insulin resistance are the main factors of T2DM, so protecting islet β cells is beneficial to delay the occurrence of diabetes. Studies have shown that curcumin can increase glucose uptake by skeletal muscle and improve insulin resistance by inhibiting islet β-cell apoptosis [107].

Enzymes related to glucose consumption are also important targets for treating diabetes. G-6-pase and PEPCK are rate-limiting enzymes related to liver gluconeogenesis, and enhanced activities of these enzymes lead to an increase in liver sugar output and insulin resistance in patients with type 2 diabetes mellitus [108]. The activation of carbohydrate hydrolases, such as α-glucosidase, accelerate the absorption of glucose. Curcumin can inhibit the activities of PEPCK, G-6-pase and α-glucosidase, as well as reduce the expression of G-6-pase and PEPCK proteins in mice to regulate glucose metabolism [109].

Curcumin also has curative effects on chronic complications caused by diabetes. For example, curcumin has potential to treat diabetic retinopathy by antioxidant and antiangiogenesis [110], to treat diabetic myocardial diseases by inhibiting myocardial fibrosis [111] and to treat diabetic nephropathy by reducing inflammatory reaction [112,113].

### 4.3. Anticoagulation and Antiplatelet Aggregation

Platelet activation and aggregation are recognized as core events in atherosclerotic thrombosis. The secretion and aggregation of platelets is a complex cascade activated by epinephrine, adenosine diphosphate, platelet-activating factor, collagen and arachidonic acid [114]. Curcumin may affect the cascade by directly inhibiting the deacetylation of arachidonic acid or by indirectly interfering with the binding to platelet phospholipids [115]. Moreover, curcumin can directly block the calcium signaling pathway, antagonize the GP II B/III A receptor, inhibit the cyclooxygenase pathway and hinder the formation of thromboxane A2, inhibiting platelet aggregation and thrombosis [116].

## 5. Curcumin in Treatment of Viral Infection

Since 2019, the global pandemic caused by the novel coronavirus has been escalating, with more than six million deaths to date. Although a wide range of vaccinations have been administered out around the world, the mutations of the virus are very diverse, so the development of drugs to treat COVID-19 is imminent. Recently, Paxlovid and bebtelovimab were approved for marketing, but clinical evidence is limited, and there are skeletal adverse effects, with bebtelovimab is temporarily listed as an alternative therapy in the NIH guidelines. The potential targets of COVID-19 treatment are mainly spike (S) protein blocking, angiotensin-converting enzyme 2 (ACE2) inhibitor, etc. [117]. The novel coronavirus S protein has a very high affinity with the ACE2 receptor on the human extracellular vesicle (EV), and the S-protein-neutralizing antibody blocks the opportunity for the virus to fuse with the human ACE2 by binding to the viral protein, cutting off the virus host pathway. Wu et al. [118] extracted EV-expressing ACE2 kidney cells, constructed a fake virus that carries S protein and can bind to ACE2 and found that EV can block infection by a fake virus in normal host cells; that is, too much ACE2 can competitively inhibit the binding of viral S protein to human ACE2. The team then used curcumin to stimulate kidney cells that overexpressed ACE2 and found that the EV secreted by the cells could express more ACE2. Transmissible gastroenteritis virus (TGEV) belongs to the α-coronavirus genus of the coronavirus family and has a certain homology with the novel coronavirus. Li et al. [119] treated cells with curcumin in advance and then infected cells with TGEV and found that curcumin inhibited the proliferation of TGEV and the expression of viral-related proteins in a dose-, temperature- and time-dependent manner. Due to the highly contagious nature of the novel coronavirus, it is difficult to directly use the virus as a research object, but the potential of curcumin to fight novel coronavirus has begun to emerge.

Although the effect of curcumin on the novel coronavirus is not definitive, there is evidence of inhibition of other viruses. Early studies have found that curcumin can inhibit influenza viruses, such as human respiratory syncytial virus, H1N1 and H3N3 [120]. Curcumin also has a certain inhibitory effect on common human pathogenic infectious viruses. For example, curcumin can significantly inhibit the long terminal repeat (LTR) of the human immunodeficiency virus (HIV) gene under the premise of not affecting cell activity and affect the virulence and replication ability of the virus [121]. Curcumin can also inhibit coxsackievirus by reducing viral RNA expression and protein synthesis, degrading the ubiquitin-proteasome system (UPS) and reducing the viral titer [122]. Wei et al. [123] found that after curcumin intervention, the expression levels of hepatitis HBV surface antigen (HBsAg) and hepatitis HBV e antigen (HBeAg) of the HBV stable mutant HepG2.2.15 were significantly downregulated, and the viral load of intracellular covalently closed circular DNA (cccDNA) and HBV DNA was decreased simultaneously. Curcumin can also downregulate HCV gene expression and viral replication by inducing heme oxygenase-1 expression or inhibiting the Akt-SREBP-1 pathway [124]. High-risk human papillomaviruses (HPVs) expressing E6 and E7 proteins are important factors for the occurrence and development of cervical cancer and oral cancer. A study revealed that curcumin can downregulate the nuclear transcription factors AP-1 and NF-κB and selectively inhibit the transcription of the HPV-16/E6 proto-oncogene [125].

In addition to the above common viral infectious diseases, the diseases caused by herpes virus are particularly noteworthy. The herpes virus can form a lifelong infection in the incubation period and has the ability to reactivate periodically [126]. Long-term administration of antiherpes drugs leads to various adverse effects and drug resistance. Focusing on natural compounds of plant origin is a new strategy for the treatment of herpes viruses. It has been found that curcumin exhibits excellent therapeutic potential for antiherpes virus treatment. Curcumin can affect the recruitment of IE gene promoters by RNA polymerase II mediated by viral transactivator protein vp16 through a pathway that is not dependent on the activity of transcriptional coactivator protein p300/CBP histone acetyltransferase, thereby inhibiting IE gene expression and inhibiting herpes simplex virus-1 (HSV-1) infection [127]. Curcumin may also inhibit HSV-2 replication by downregulating NF-κB. It has also been found that curcumin can inhibit other herpesviruses. For example, curcumin may inhibit human cytomegalovirus by downregulating IEA, UL83A and Hsp90 [128], as well as Kaposi’s sarcoma-associated herpesvirus by blocking APE1-mediated redox function [129] and the pseudorabies virus (PRV) via transcription of BZLF1 against EBV virus [130].

## 6. Hurdles in the Applications of Curcumin

Curcumin displays considerable potential in the management of human pathological conditions; meanwhile, hurdles in the clinical practice of this compound are equally impressive.

### 6.1. Low Bioavailability and Solubility

Curcumin exhibits extremely poor water solubility, low absorption in the small intestine and rapid elimination in the liver. Animal experiments showed that after ingestion by rats, most curcumin was excreted in feces in the prototype. Human studies have shown that the peak plasma concentration of curcumin is only 11.1 nmol/L 1–2 h after intake of 3.6 g curcumin [131]. Therefore, the low bioavailability of curcumin considerably limits its clinical applications, and efforts are invested to improve the bioavailability.

#### 6.1.1. Combination with Other Drugs

Piperine can increase the absorption of curcumin, inhibit its glucuronidation and improve its bioavailability by more than 20 times [1]. Lycopene can increase the antioxidant function of curcumin and effectively improve its antioxidation in mice with acute ethanol oxidative injury [132]. Glycyrrhetinic acid combined with curcumin could inhibit the proliferation of liver cancer cells and induce apoptosis more effectively [133]. Metal ions, such as Zn^2+^, Cu^2+^, Mg^2+^ and Se^2+^, as well as serum albumin, can form complexes with curcumin to improve its water solubility and bioavailability. These findings shed light on the clinical translation of curcumin.

#### 6.1.2. Curcumin Complex

In this section, we summarize the curcumin-embedding complexes with liposomes, polysaccharides or protein, cyclodextrin, nanoparticles and other carriers developed over the past few years and describe them as follows in order to provide a theoretical basis for the optimization of the pharmacological characteristics of curcumin.

Liposomes are emerging nanopharmaceuticals composed of a lipid bilayer (phospholipid bilayer and cholesterol) surrounding an internal water environment, which can embed both lipophilic and hydrophilic compounds to improve solubility and permeability, thus improving the bioavailability [134]. Studies have shown that curcumin liposomes based on phospholipids, cholesterol and Tween-80 can improve the stability of various pH values and metal ions, in addition to enhancing cell absorption and antioxidant properties [135]. Curcumin liposome based on lecithin can prolong the plasma half-life. Curcumin liposomes using a combination of phospholipids, sodium hydroxide and ethanol can also improve the stability of biological storage [136]. Curcumin liposomes based on soy lecithin, nonionic surfactants and cholesterol can slow the rate of sustained curcumin release [137]. Borneol-modified curcumin cationic liposomes can significantly increase the contents of curcumin in vivo and in brain tissues and delay elimination after nasal administration [138].

In addition to the liposome, curcumin can also be combined with polysaccharides or biocompatible proteins with low immunogenicity; this type of biopolymerization nanoparticle can be trapped in the granules by hydrophobic and hydrogen bonding by inducing cell endocytosis, on the one hand, or enhance the direct absorption by the small intestinal epithelial cells, on the other hand, to inhibit curcumin in the intestine oxidation decomposition reaction, enhancing the solubility and bioavailability of curcumin. For example, chitosan-curcumin has better antioxidant properties with metal chelating properties [139]; acetylated starch-curcumin complex has better sustained-release capacity [140]; and quercetagetin-curcumin [141], silk fibroin-curcumin [142], *Pseudostellariae* protein-curcumin [143], rice bran waste curcumin, etc., can improve the water solubility [144], photosensitivity and controlled release of curcumin to varying degrees.

Cyclodextrins are starch-derived oligosaccharides with an internal hydrophobic and hydrophilic surface structure, which enhances the solubility of hydrophobic compounds in water [145]. Curcumin precursor liposomes can reduce the defect of poor stability and improve bioavailability in rats by more than 10 times [146]. Β-cyclodextrin increases the solubility of curcumin by forming a 1:1 type host–guest complex with curcumin [147]. In vitro experiments have also proven that curcumin encapsulated with cyclodextrin may produce higher blood concentrations of curcumin [148]. Curcumin was also dispersed in a high-molecular-carrier material to form a microsphere dispersion and a complex with alicyclic acid at a ratio of 1:1, thus improving the bioavailability and blood concentrations [149]. A curcumin-selenium complex was designed have a better therapeutic effect on cadmium-induced liver injury in rats [150]. Drug eutectics formed by the compatibility of Chinese medicine, such as curcumin-artemisinin eutectic [151] and curcumin-ginseng ketone IIA eutectic [152], can also improve the bioavailability of curcumin.

Encapsulating curcumin in nanoformulations is another important strategy to improve its bioavailability, whereas the toxicity of curcumin to non-targeted tissues is reduced [153]. Compared with curcumin, the content of curcumin in curcumin nanoparticles wrapped in poly(lactic-co-glycolic acid) increased by nine times [154], and the anti-HIV activity of curcumin nanoparticles increased by three times [155].

### 6.2. Controversial in Purity

The total extract of turmeric contains curcumin and other unknown components, such as methoxycurcumin and bisdemethoxycurcumin [1]. Xu et al. [156] examined three curcumin compounds and volatile components in 160 turmeric samples from five provinces of China, and the results showed that the curcumin contents in 26% of the samples did not meet the standards of the Chinese Pharmacopoeia. It is noteworthy that the purity of curcumin is not necessarily proportional to the effect. Ren et al. reported that pure curcumin did not exhibit anticoagulant function in the thrombus model of rats, whereas the total extract of turmeric showed a strong anticoagulant effect [157]. Studies also showed that the total extracts of turmeric, curcumin, demethoxycurcumin and demethoxycurcumin all have certain inhibitory effects on the proliferation of human lung adenocarcinoma cells A549, colon cancer cells HT29 and glioblastoma cells T98G, but the total extracts of turmeric has the strongest activity [158]. The deficiency of curcumin contents in Curcuma longa may directly affect the quality of curcumin extracts, but the pharmacological effects of non-curcuminoid parts cannot be ignored.

Curcumin (or curcuminoid) is favored by consumers as a natural dietary supplement. A report on curcumin contents of turmeric dietary supplements in American city retail markets showed that 71% of the products contain solvent residues that meet the standard but are unnecessary components in the curcumin products. It was also found that 59% of curcumin products did not meet the standard of curcumin contents at 80%, which may be related to the use of unknown synthetic curcumin. Therefore, there may be differences in the formulation of curcumin content standards in the fields of food and medicine.

### 6.3. Lack of Supported Clinical Reaserch Results

The search term “curcumin” on the website “National Institutes of Health Clinical trial.gov” returned 288 clinical studies; 140 clinical studies were listed as completed, but only 20 had results available. Of the 288 clinical studies, 17 were associated with curcumin with respect to cancer and 1 with abnormal blood conditions. Unfortunately, there were no clinical studies of AD or viral infections, and only 3 out of 18 had findings available. Basic information about these 18 clinical studies is presented in Table 1 in the order in which the studies were recruited. According to the information in the table, although curcumin has not disappeared from the literature for 20 years, compared with the preclinical study of curcumin, its clinical study is not satisfactory for reasons such as small sample size, focus on a single disease type (mainly breast cancer, colorectal cancer, prostate cancer, etc.), the difference between curcumin interventions (mainly dose, dosage form and medication time) and an excessive scope of the study endpoints (such as survival, safety, tolerance, etc.). Most importantly, the finding of few of these clinical studies have been published. This shows that curcumin still has a long way to go to practical clinical application.

## 7. Side Effects of Curcumin

Centuries of life experience of taking curcumin through daily diet seems to give people considerable confidence in its safety. Curcumin has become one of the world’s best-selling dietary supplements, with an increasing number of consumers favoring the ingredients extracted from natural plants. An in vivo experiment showed that there were no adverse effects when dogs or monkeys were given 3500 mg/kg curcumin orally [159]. A clinical study involved patients with rheumatoid arthritis taking 1200–2100 mg curcumin orally every day for 6 weeks without adverse effects [160]. Additionally, intervention with curcumin in advanced colon cancer showed that administration of up to 3600 mg daily for four months was well tolerated as a whole, with diarrhea and nausea in some cases, although these side effects were relieved by themselves [161]. The Joint FAO/WHO Committee of Experts on Food Additives (JECFA) and the European Union Food Science Committee (SCF) have developed a recommended daily intake of 0–3 mg/kg curcumin. Although the medical community believes that curcumin in food has only a low toxicity and does not pose a threat to human beings or the environment, we cannot take it for granted that curcumin is absolutely safe; that is, the side effects caused by high doses of curcumin cannot be ruled out. We summarize the possible side effects of high-dose curcumin in the following sections (Table 1), although the majority of these events were isolated cases. Extensive evaluation is warranted.

### 7.1. Risk of Kidney Calculi in Susceptible Individuals

Urinary oxalate excretion over 40 mg/24 h is called hyperoxaluria, which is one of the main risks of kidney calculi. The content of oxalate in urine depends on endogenous synthesis and absorption of exogenous oxalate. The absorption rate of oxalate in different foods is variable, at about 2–15%, and the higher the solubility of oxalate, the higher the absorption rate. A study showed that a supplementary dose of turmeric (91% oxalate soluble) or cinnamon (6%) for 4 weeks was equivalent to 55 mg/day oxalate [162]. The oxalate loading test indicated that the oxalate absorption rate of turmeric (8.2%) in 6 h was significantly higher than that of cinnamon (2.6%), and thus, the turmeric group exhibited significantly higher urinary oxalate excretion compared to the cinnamon group. In kidney calculi, the rate of endogenous oxalate synthesis increases, and more exogenous oxalate is absorbed [163]. Therefore, excessive consumption of turmeric may increase the urinary oxalate level, as well as the risk of kidney calculi formation in susceptible individuals.

### 7.2. Anemia in Patients with Iron Failure

Curcumin can mediate the reduction in H and L subunits of hepatocyte ferritin, increase transferrin receptor 1 (TfR1) expression and inhibit the synthesis of systemic hepcidin [164]. Therefore, curcumin might have similar properties as an iron chelator, inducing iron failure [165]. After mice were fed a diet containing different concentrations of iron (5, 12, 50 and 1000 mg/kg) and curcumin (0, 0.2, 0.5, and 2.0%) for 26 weeks, the mice that received the lowest level of iron showed critical iron deficiency (i.e., decreased H and L subunits of protein and increased TfR1) and decreased spleen and liver iron, but these mice exhibited only slight changes in phenotypic parameters of iron-deficient anemia, such as hemoglobin, hematocrit, transferrin saturation and serum iron. However, in the context of subclinical iron deficiency (iron intake < 1000 mg/kg), curcumin intake induced a concentration-dependent decrease in these blood parameters [166]. Therefore, high-dose curcumin should be used cautiously in patients with subclinical iron deficiency, chronic anemia and consumptive anemia caused by excessive loads of malignant tumors, although the antitumor effects may be boosted by curcumin-induced iron failure.

### 7.3. Liver Injury

It has been reported that a 78-year-old healthy woman developed jaundice symptoms and was diagnosed with hepatic injury hepatitis after she took a curcumin supplement at a high dose (500 mg/d) for one month. The peak values of aspartate transaminase (AST) and alanine aminotransferase (ALT) in laboratory tests were 20 times higher than the normal upper limit, but the AST and ALT levels decreased by 48% on the 7th day after curcumin supplementation was stopped, and transaminase returned to normal on the 42nd day [167]. Lukefahr et al. [168] also reported a case of autoimmune hepatitis induced by drugs containing curcumin supplements. Although there are not many reports on liver injury caused by curcumin, the existing reports emphasize the importance of curcumin supplementation as a triggering factor of drug-induced liver injury.

### 7.4. Abnormal Cardiac Conductions

It was reported that a 38-year-old healthy male suddenly felt dizzy, with chest tightness, nausea and sweating after he took curcumin supplements at a high dose (1500 mg/d) for one month. An electrocardiogram showed complete atrioventricular blockage, and there were countless asymptomatic second-degree and Mohs type I atrioventricular blockages in one night. After 3 days of drug withdrawal, the patient’s ECG recovered to sinus. Causative investigations revealed that the same symptoms recurred after reconsumption of the same amount of the curcumin supplements [169].

### 7.5. Influence on Other Drugs

Tamoxifen is an endocrine therapeutic drug that downregulates the estrogen receptor signaling of breast cancer cells. In the human body, tamoxifen is metabolized into active edoxifen. Researchers genotyped and grouped the recruited breast cancer patients into high expression of CYP2D6 vs. normal expression and then administered tamoxifen treatment (20–30 mg, qd) alone compared to in combination with curcumin (1200 mg, tid) [170]. The results showed that compared to the single-drug tamoxifen treatment, the AUC_0-24h_ of the edoxifen and tamoxifen decreased by 7.70% and 12.98%, respectively, in the curcumin combination group; furthermore, the blood concentrations of patients with high expression of CYP2D6 decreased to a much lower degree (10.3% vs. 7.2%). Therefore, especially in patients with high-expression CYP2D6, the use of curcumin significantly reduces the pharmacokinetic parameters of tamoxifen and edoxifen, and patients may not be able to undergo optimal endocrine therapy. In order to ensure the efficacy of tamoxifen, the treatment of curcumin should be stopped as soon as possible in breast cancer patients.

Cyclophosphamide is a commonly used cytotoxic chemotherapy drug in clinical settings. After the cytotoxic drug cyclophosphamide was intraperitoneally injected into rats pretreated with curcumin seven days prior, the proper chromosome arrangement and mitotic index were significantly decreased [171]. Therefore, the antidistortion effect of curcumin may weaken the antitumor effect of cyclophosphamide on tumor cells.

### 7.6. Allergic Reactions

Curcumin-induced contact urticaria may be mediated by immune- and non-immune-mediated mechanisms. Immune-mediated urticaria usually goes beyond the contact and spreads throughout the body, whereas non-immune-mediated lesions are confined to the contact areas. It has been reported in the literature that a female worker in a pharmaceutical factory without allergic history exhibited skin pruritus after initial contact with curcumin. After 30 min, she found a large number of erythema scales on her head, neck and distal arm, with no involvement in the non-exposed parts. A curcumin needling test was positive [172]. Moreover, a miller developed delayed contact dermatitis after grinding turmeric powder, as did an elderly woman after use of turmeric powder dissolved in coconut oil for leg massage, both of whom presented with pruritus, erythema, papules and vesicles at the contact sites, with a positive curcumin patch test [173]. It is possible that curcumin was sensitized by reacting with dust in the air during the grinding process, resulting in contact dermatitis. However, further case and mechanistic studies are warranted.

### 7.7. Cancer Induction

Many studies have shown that the oxidative stress of curcumin occurs before the antioxidant effect, which may be related to time or dose dependence [174]. Therefore, the oxidative induction of curcumin may promote tumorigenesis under certain specific conditions. Doxycycline (DOX) was used to induce progression-free benign lung lesions in rats to mimic a population with a history of smoking and damaged lung epithelial cells but no malignant lung lesions. Research revealed that compared with simple DOX treatment, DOX combined with curcumin, DOX combined with butylated hydroxytoluene (BHT, a lung cancer inducer) and DOX combined with curcumin and BHT all could lead to tumor multiplicity and hyperplasia development, with increased recruitment of leukocytes to tumor cells and inflammation, thus leading to increased oxidative stress in the tumor microenvironment [175]. This study showed that curcumin might act as a prooxidant to induce tumor progression in the hyperoxia environment of the lung, i.e., early benign lung lesions. Similarly, a clinical trial of curcumin intervention in high-risk groups of malignant tumors showed that three months of curcumin treatment (doses from 500 to 8000 mg/d) resulted in aggravation of histological precancerous lesions in individual patients after bladder cancer surgery, cervical intraepithelial neoplasia, intestinal metaplasia and Bowen’s disease [176]. In summary, curcumin may not be suitable for people at high risk of cancer.

### 7.8. Other Risks

The effects of curcumin on blood sugar and platelet aggregation have been documented. Although no literature reports the risk of hypoglycemia and bleeding when curcumin is used in combination, studies have shown that curcumin may delay the metabolism of glibenclamide by inhibiting the activity of CYP3A, which has a synergistic effect, along with glibenclamide, lowering blood sugar in type II diabetic rats [177]. Two clinical studies on curcumin intervention in AD conducted by Baum [178] and Ringman [179] excluded people who were treated with anticoagulant or antiplatelet drugs or had bleeding risk factors. Therefore, in the use of curcumin, one should avoid combination medications to the greatest extent possible in order to prevent hypoglycemia and bleeding. In vitro experiments also show that 300 μg/mL curcumin can thoroughly inhibit the mobility of healthy human sperm [180]. In addition, previous studies have shown that curcumin has an antiviral effect on HIV and is therefore considered to have the potential for the development of vaginal contraceptives. However, whether the lethality of curcumin against sperm affects healthy people during pregnancy still needs to be treated with caution.

## 8. Concluding Remarks

Plant-derived curcumin is still a regularly found at Asian dining tables, and its human health benefits have been confirmed through pharmacological research over the past three decades. Curcumin seems to be a panacea due to its multitarget and multichannel characteristics, displaying a wide range of pharmacological properties in the fields of anti-malignant tumor treatment, AD treatment, blood lipid and blood glucose control, anti-coagulation, antiplatelet, antiviral treatment, etc. What is shocking is that curcumin can theoretically inhibit the novel coronavirus. However, compared with a large number of exciting preclinical studies on curcumin, the clinical studies on curcumin are rather dull. No clinical study has clearly indicated that curcumin can exactly inhibit the progression of malignant tumors, improve cognitive function in AD, treat metabolic syndrome, block virus infection, etc. Curcumin sold on the market exists in the form of healthcare products or dietary supplements, with the aim of relieving alcoholism, protecting the liver, preventing inflammation and relieving pain—in other words, it has little to do with the treatment of diseases.

In recent years, focusing the defects of low water solubility and bioavailability of curcumin, an increasing number of curcumin complexes have been identified as being combined with compounds, embedded in special materials, etc., which theoretically enhance the bioavailability of curcumin and even enhance its pharmacological effects. This also clarifies the fact that low-purity curcumin is not synonymous with low efficacy. However, few of these curcumin complexes have been truly reflected in the actual application with respect to human health. The safety and toxicity of the compounds and special materials used to enhance bioavailability should be considered due to the excessive use of additives and colorants harmful to human health.

Although numerous in vivo experiments have demonstrated that curcumin at the prescribed dose does not have side effects in the body, we cannot take it for granted that curcumin is absolutely safe. Individual case reports of adverse effects manifested in the heart, liver, kidney, blood, reproduction and immune system cannot be ignored, and the fact that curcumin may have a cancer-promoting effect on populations with high risk factors for cancer or precancerous lesions requires careful attention. In conclusion, the potential benefits of curcumin for human health are enormous. However, clinical translational application and adverse effects management may pose the greatest challenge for curcumin.

## Figures and Tables

**Figure 1 molecules-27-04400-f001:**
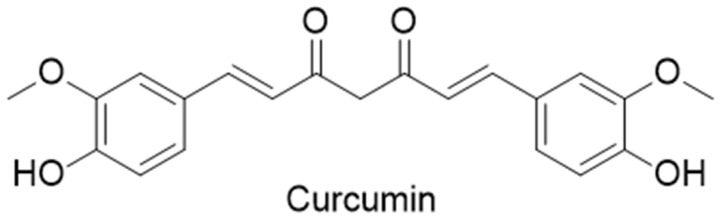
Chemical structure of curcumin.

**Figure 2 molecules-27-04400-f002:**
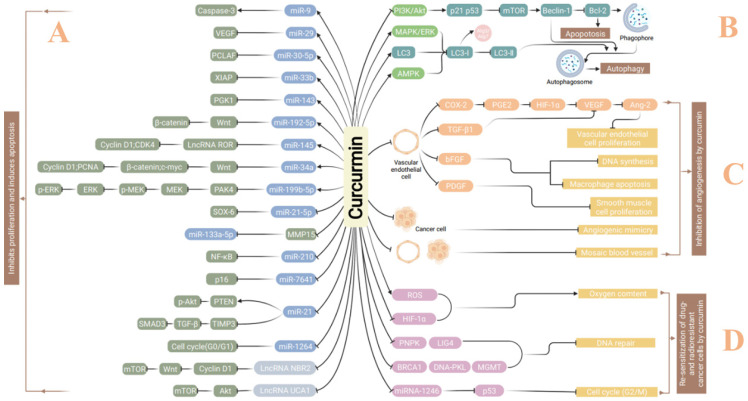
Part A: Curcumin inhibits proliferation and induces apoptosis of cancer cells through miRNA and lncRNA-mediated mechanisms. Part B: Curcumin-mediated autophagy of cancer cells. Curcumin may inhibit the PI3K/Akt signaling pathway to activate the autophagy marker Beclin-1, which induces apoptosis via inhibition of Bcl-2 and promotes the formation of autophagosomes from the free phagophores, inducing autophagy. Curcumin can also activate the MAPK/ERK and AMPK signaling pathways or LC3 to induce autophagy via an LC3-II-mediated mechanism. Part C: Inhibition of angiogenesis by curcumin. Curcumin can inhibit the proliferation of vascular endothelial cells and smooth muscle cells through COX-2/PGE2, TGF-β1-, bFGF- and PDGF-mediated mechanisms. Curcumin can also inhibit the angiogenic mimicry of cancer cells and mosaic blood vessel formation in co-cultures of cancer cells and vascular endothelial cells. Part D: Re-sensitization of radioresistant cancer cells by curcumin. Curcumin can increase oxygen content through upregulation of ROS or downregulation of HIF-1α, inhibit DNA repair and promote cancer cell arrest at G2/M through the miRNA-1246-mediated p53 pathway.

**Figure 3 molecules-27-04400-f003:**
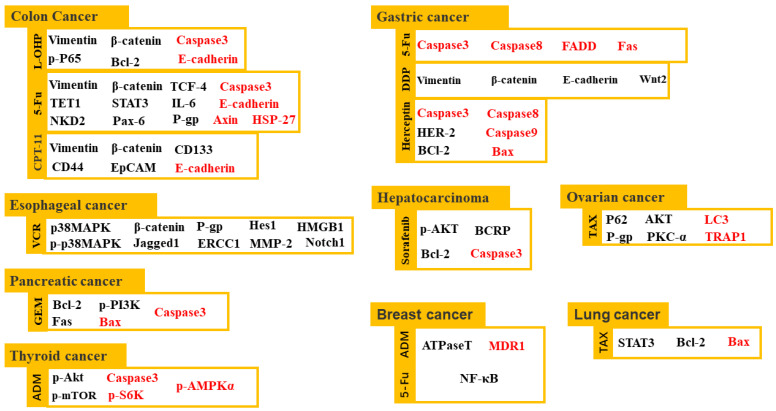
Curcumin reverses chemotherapeutic resistance of multiple cancer cell lines through regulation of multiple gene expression. The black words represent the targets downregulated by curcumin, whereas the red words denote the targets upregulated by curcumin. (L-OHP, oxaliplatin; 5-Fu, 5-fluorouracil; CPT-11, irinotecan; VCR, vincristine; GEM, gemcitabine; ADM, doxorubicin; DDP, cisplatin; and TAX, paclitaxel).

**Table 1 molecules-27-04400-t001:** Details of curcumin clinical studies.

Number	Status	ID	Phase	Study Start Date	Study Completion Date	Sponsor	Condition or Disease	Number of Participants	Intervention/Treatment	Research Endpoint
1	Completed	NCT00192842	2	Jul-04	Sep-10	Rambam Health Care Campus	Advanced Pancreatic Cancer	17	Curcumin 8 g po qd (+gemcitabine)	To assess the efficacy of the standard chemotherapy gemcitabine plus curcumin in patients with advanced pancreatic cancer.
2	Completed	NCT00094445	2	Nov-04	Apr-14	M.D. Anderson Cancer Center	Advanced Pancreatic Cancer	50	Curcumin 8 g po qd for 22 weeks	To learn if treatment with curcumin can help slow the growth of pancreatic cancers and to access the safety.
3	Completed	NCT00113841	-	Nov-04	Aug-09	M.D. Anderson Cancer Center	Patients With Multiple Myeloma	42	Curcumin 2 g po bid (+Bioperine 5 mg po bid)	To assess the Quality of life (QOL).
4	Completed	NCT03211104	-	Aug-07	Aug-15	Samsung Medical Center	Patients With Prostate Cancer Undergoing Intermittent Androgen Deprivation Therapy	107	Curcumin 240 mg po tid for 6 month	To assess whether curcumin influences the duration of treatment interruption and rate of prostatic specific antigen (PSA) progression.
5	Recruiting	NCT00745134	2	Aug-08	Mar-23	M.D. Anderson Cancer Center	Rectal Cancer	45	Curcumin po bid +(Radiation therapy and capecitabine) for 11.5 weeks	To asscess if curcumin can make tumor cells more sensitive to radiation therapy.
6	Completed	NCT01160302	1	Jun-10	Jan-16	Louisiana State University Health Sciences Center Shreveport	Head and Neck Cancer	33	Microgranular Curcumin C3 Complex^®^ 4 g po bid for 21–28 days	To assess the adverse effects.
7	Completed	NCT01333917	1	Nov-10	Jan-13	University of North Carolina, Chapel Hill	People with positive colonoscopy screening	40	Curcumin C3 4 g qd for 30 days	To identify genes that are modified by curcumin that could be used as biomarkers in future chemoprevention studies, and also to evaluate the tolerability and toxicity.
8	Completed	NCT01917890	-	Mar-11	Oct-13	Shahid Beheshti University of Medical Sciences	Prostate Cancer	40	Curcumin or placebo (500 mg po qd for 7–8 weeks)	To assess Progression free survival, Time to Disease Progression and Time to treatment failure.
9	Completed	NCT01490996	2	Feb-12	May-17	University of Leicester	Inoperable Colorectal Cancer	41	Curcumin C3 2 g po qd (+FOLFOX)	To assess the safety, tolerability, efficacy(measured by response rate with RECIST and overall survival in months).
10	Completed	NCT01740323	2	May-15	Jul-18	Emory University	Chemotherapy-Treated Breast Cancer Patients Undergoing Radiotherapy	30	Curcumin or placebo (500 mg po bid)	To assess if curcumin reduces NF-kB DNA binding and its downstream mediator IL-6.
11	Completed	NCT02439385	2	Aug-15	Aug-19	Gachon University Gil Medical Center	Colorectal Cancer Patients With Unresectable Metastasis	44	Curcumin 100mg po bid (+Avastin/FOLFIRI)	To assess Progression free survival, Overall survival rate, Overall response rate, safety and fatigue score.
12	Completed	NCT02321293	1	Aug-15	Dec-16	Lady Davis Institute	EGFR -Mutant Advanced NSCLC	20	Longvida® Optimized Curcumin 80 mg po qd (+Tyrosine Kinase Inhibitor) last for 8 weeks	To assess the safety and tolerability.
13	Recruiting	NCT02724202	1	Mar-16	-	Baylor Research Institute	Colon Cancer	13	Curcumin 500 mg po bid for 2 weeks. (Patients will continue on curcumin at same dose for an additional 6 weeks while being treated with 3 cycles of 5-Fu)	To test the safety, effects and find the Response Rate.
14	Completed	NCT03072992	2	Mar-17	Jun-19	National Center of Oncology, Armenia	Advanced Breast Cancer	150	Paclitaxel +(curcumin or placebo) (300 mg i.v. once weekly for 12 weeks.)	To assess the adverse effects, Quality of life, Progression free survival, Time to Disease Progression, and Time to treatment failure.
15	Completed	NCT03534024	-	Aug-18	-	National Nutrition and Food Technology Institute	Metabolic Syndrome	50	Nanomicielle curcumin or placebo	To determine the effects of nanomicelle curcumin on glycemic control, serum lipid profile, blood pressure and anthropometric measurements
16	Recruiting	NCT03769766	3	Mar-19	-	University of Texas Southwestern Medical Center	Prostate Cancer	291	Curcumin or placebo (500 mg po bid)	To assess the efficacy.
17	Recruiting	NCT03980509	1	Jan-20	-	Medical University of South Carolina	Invasive Breast Cancer	20	Curcumin 500 mg po bid. (Curcumin will be given from the time surgical resection is scheduled until the night before surgical resection.)	To determine whether curcumin causes biological changes in primary tumors of breast cancer patients.
18	Not yet recruiting	NCT04294836	2	Dec-21	-	Instituto Nacional de Cancerologia, Columbia	Cervical Cancer	240	Cisplatin plus concomitant radiation therapy (teletherapy + high or low rate brachytherapy) + Curcugreen (BCM95) or placebo 2000 mg daily (each 6 h)	To assess the efficacy and safety.

## Data Availability

Not applicable.

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
