# Peer review of "A Comprehensive Review on the Benefits and Problems of Curcumin with Respect to Human Health"

_molecules, 2022, doi:10.3390/molecules27144400_

Round 1
Reviewer 1 Report
Title: "Curcumin: An enigma in medicine”
Authors: Si-yu Liu, Liu Liu, Lan He, Jie Liu, Fang-liang Zhou, Bo Cheng,
De-liang Cao, Ying-chun He
Comments:
In this review article, the authors discuss the existing hurdles and side effects of curcumin as a medicinal agent and provide suggestions for basic research and clinical application of this agent.
Several Points:
General:
1: The topic of the review is very interesting, as it also considers critical aspects of treatment with curcumin. The writing style is rough (lack of transitions, little own thoughts) and lacks tables and very important references. For this research have some more questions and suggestions, details are as follows:
2: Chapters 2.1-2.5: For these chapters, a tabular listing with cancer type, target, effect, and reference would be helpful.
3: Chapter 2.2, here I suggest changing the sentence order to make the abbreviation understandable: Kim et al. used curcumin to treat cells from oral squamous cell carcinoma (OSCC).
4: Chapter 2.3: Please check the text position "Zhang et al. (49)" as reference 49 should be Yin et al. according to the list.
5: Chapter 2.3: In this chapter, the importance of EMT is highlighted. Please note that EMT is promoted by intercellular cross-talk in the tumor microenvironment and that curcumin multifactorially counteracts this according to reference (DOI: 10.1016/j.bcp.2006.12.010; doi: 10.1371/journal.pone.0107514; doi: 10.1002/ijc.24593).
6: Chapter 2.4: Please check text passage "Li's group (53)" as reference 53 should be Fang et al. according to the list.
7: Chapter 2.5: Here, in addition to cisplatin, a paragraph could also be devoted to the chemotherapeutic agent 5-FU, since it is frequently used and many findings are known in connection with curcumin, please add the following references: doi: 10.1002/mnfr.201100270.; doi: 10.3892/ijo.2022 .5375; doi: 10.1186/s12885-015-1291-0; doi: 10.1007/s11356-022-20208-y; doi: 10.1371/journal.pone.0085397; doi: 10.3390/ijms23042144.
8: Chapter 2.5/3: I miss a transition between the topics of cancer and Alzheimer's disease, as the connection does not seem logical to the reader, but like a big topic jump.
9: Chapter 4.1, first sentence: please check what is meant here and if there is an error in the text! A high LDL level is harmful, while a high HDL level is beneficial to health, with a few genetic exceptions.
10: In this context, please also check reference 90 (Qin S.) in the reference list. Can this master's thesis perhaps be replaced by the author's published review (doi: 10.1186/s12937-017-0293-y)?
11: Chapter 6: A table with side effect, study type (e.g. single case), year and reference would be very helpful.
12: Figure 2: The quality of the figure needs to be improved. A larger figure and a different font color would lead to better readability. Also, it would be useful to use letters to indicate the areas described in the legend to make them easier to find.
13: Figure 3: Caspase is misspelled in several places and should be corrected. The font should also be standardized.
14: English, style, and formatting throughout the paper should be revised. There should be smooth transitions, varied sentence beginnings, more of the author's own thoughts, and minor formatting adjustments (spaces, parentheses, spelling, etc.).
Author Response
1:Chapters 2.1-2.5: For these chapters, a tabular listing with cancer type, target, effect, and reference would be helpful.
Reply:Dear reviewer, Chapters 2.1-2.5 are shown in Figure2. Considering that two tables have been added to this article in new revision version, in my humble opinion ,this section may be better presented as a picture.
2: Chapter 2.2, here I suggest changing the sentence order to make the abbreviation understandable: Kim et al. used curcumin to treat cells from oral squamous cell carcinoma (OSCC).
Reply:Dear reviewer, thank you for your suggestion. It has been revised according your suggestion in new revision version.
3: Chapter 2.3: Please check the text position "Zhang et al. (49)" as reference 49 should be Yin et al. according to the list.
Reply:Sorry, there is a mistake here. After verification, it has been revised
4: Chapter 2.3: In this chapter, the importance of EMT is highlighted. Please note that EMT is promoted by intercellular cross-talk in the tumor microenvironment and that curcumin multifactorially counteracts this according to reference (DOI: 10.1016/j.bcp.2006.12.010;doi:10.1371/journal.pone.0107514;doi:10.1002/ijc.24593).
Reply:Dear reviewer, thank you for your suggestion. I added curcumin to reverse the crosstalk between EMT and tumor microenvironment. Unfortunately, it was verified that “DOI: 10.1016/j.bcp.2006.12.010” had been retracted, so this document was not added, but two other documents were added in new revision version.
5: Chapter 2.4: Please check text passage "Li's group (53)" as reference 53 should be Fang et al. according to the list.
Reply:Dear reviewer, sorry, there is a mistake here. After verification, it has been revised.
6: Chapter 2.5: Here, in addition to cisplatin, a paragraph could also be devoted to the chemotherapeutic agent 5-FU, since it is frequently used and many findings are known in connection with curcumin, please add the following references: doi: 10.1002/mnfr.201100270.; doi: 10.3892/ijo.2022 .5375; doi: 10.1186/s12885-015-1291-0; doi: 10.1007/s11356-022-20208-y; doi: 10.1371/journal.pone.0085397; doi: 10.3390/ijms23042144.
Reply:Dear reviewer, thank you for your suggestion and the above articles have been added to Chapter 2.5 in new revision version.
7: Chapter 2.5/3: I miss a transition between the topics of cancer and Alzheimer's disease, as the connection does not seem logical to the reader, but like a big topic jump.
Reply:Dear reviewer, the subtle relationship between Alzheimer's disease and cancer is added here, as follows:
In the United States, Alzheimer's disease (AD) is the third most expensive disease after malignant tumors and cardiovascular diseases. Interestingly, studies have shown that the rate of memory decline of patients with malignant tumors before and after diagnosis is lower than that of healthy people. And the diagnosis of malignancy is associated with an 11% reduction in the incidence of alzheimer' s disease. It is possible that the inhibition of Peptideyl-Prolyl cis-transisomerase (PIN1), which is responsible for the structural regulation of protein, hyperphosphorylated TAU and enhanced the risk of AD. The polymorphism associated with the reduced expression of PIN1 has been associated with a reduced risk of cancers. Malignant tumors and AD may be natural enemies of each other. However, with the global aging, AD will be a greater challenge to all mankind than malignant tumors someday.
8: Chapter 4.1, first sentence: please check what is meant here and if there is an error in the text! A high LDL level is harmful, while a high HDL level is beneficial to health, with a few genetic exceptions.
Reply:Dear reviewer,sorry, there is a mistake here. It is revised as follows.
Abnormal blood lipids, especially the increase of low-density lipoprotein (LDL), are recognized as the pathological factors of coronary atherosclerosis
9: In this context, please also check reference 90 (Qin S.) in the reference list. Can this master's thesis perhaps be replaced by the author's published review (doi: 10.1186/s12937-017-0293-y)?
Reply: Dear reviewer, thank you for your suggestion that the articles for this study are indeed more approriate than the master thesis as a reference in my review. References have been replaced in new revision version.
10: Chapter 6: A table with side effect, study type (e.g. single case), year and reference would be very helpful.
Reply: Dear reviewer, thank you for your suggestion. The form of case report has been added to the review, which makes the description clearer.
11: Figure 2: The quality of the figure needs to be improved. A larger figure and a different font color would lead to better readability. Also, it would be useful to use letters to indicate the areas described in the legend to make them easier to find.
Reply: Dear reviewer, thank you for your suggestion, Corresponding changes have been made.
12: Figure 3: Caspase is misspelled in several places and should be corrected. The font should also be standardized.
Reply: Dear reviewer, thank you for your suggestion .The spelling of caspase in the paper has been revised.
.
13: English, style, and formatting throughout the paper should be revised. There should be smooth transitions, varied sentence beginnings, more of the author's own thoughts, and minor formatting adjustments (spaces, parentheses, spelling, etc.).
Reply: Dear reviewer, thank you for your suggestion, I have added the cohesion of different content parts in the article, making the topic conversion of the article smoother. And I am very sorry, because English is not the my mother tongue, my English style of this article is rough, our team decided to use the English polish service under MDPI to improve my poor English expression.

Reviewer 2 Report
Authors have investigated and attempted to review on curcumin as a biologically active agent in malignant tumours, Alzheimer's disease, and haematological disorders. Although the topic is interesting, it is doubtful that a comprehensive long review of curcumin's activity is needed after Augustine and colleagues provided it in 2017 (PMID: 28417091). It would be more fascinating to report on recent developments from 2017 or to re-examine the literature with a fresh and new perspective. This review fails to do so, and as a result, I am not really enthusiastic about it.
1. The objective and significance of this review should be expressed more clearly in the abstract. The abstract part is not appropriate. In the abstract portion, there is no focus point. There was no conclusion, and the abstract did not offer any future directions.
2. The authors need to make significant changes to the introduction. Include any previous research constraints, as well as how the current study addresses those limitations. In the introduction section, the study's gaps should be explicitly outlined, along with any relevant references. The introduction's flow isn't ideal, and it's vague.
3. In general, the manuscript's novelty and original potential in comparison to the published literature should be described in greater detail in the manuscript. Otherwise, it's a rehash of the curcumin review that was already published.
4. Provide the main mechanism of action/metabolic effects of curcumin in mentioned disorders; you should explain more details on the pathways, enzymes, and markers involved in the articles in the new table for quick review and better comparison of reported studies.
5. Given the wide range of effects reported, it is better to focus on newer studies and of course in clinical studies than pre-clinical studies. Please add the section to the explanation of performed recent clinical trials of Curcumin.
6. Accurate revision to the text by a native English speaker is required to resolve grammatical and syntax errors and typos.
Author Response
- The objective and significance of this review should be expressed more clearly in the abstract. The abstract part is not appropriate. In the abstract portion, there is no focus point. There was no conclusion, and the abstract did not offer any future directions.
Reply: Dear reviewer, Thank you for your suggestion, the abstract has been revised as follows:
Curcumin is the most important active component in the turmeric extracts. Curcumin, a natural monomer from plants has received a great attention as its dietary supplement feature and evident activity in a wide range of human pathological conditions. In general, curcumin is beneficial to human health, demonstrating pharmacological activities of anti-inflammation, anti-oxidation, anti-tumor, and immune regulation. Curcumin also presents therapeutic potentials in neurodegenerative diseases and cardiovascular and cerebrovascular diseases. This review article summarizes the updates achieved in recent years of curcumin as a biologically active agent in malignant tumors, Alzheimer's disease, hematological diseases and Viral infection diseases.
In addition, this paper also focuses on problems in curcumin from basic research to clinical translation, such as the low solubility of curcumin leads to poor bioavailability, and the controversy between curcumin purity and effect. At the same time, through reviewing and summarizing the clinical research of curcumin and case reports of adverse effects, We found that the clinical transformation of curcumin is not successful, and excessive intake of curcumin may have adverse effects on the kidneys, heart, liver, blood and immune system, which brings us a warning that curcumin has a long way to go from basic research to application transformation.
2.The authors need to make significant changes to the introduction. Include any previous research constraints, as well as how the current study addresses those limitations. In the introduction section, the study's gaps should be explicitly outlined, along with any relevant references. The introduction's flow isn't ideal, and it's vague.
Reply: Dear reviewer, thank you for your suggestion ,he introduction has been revised as follows:
The turmeric has a long history. Early in the 12th century, Kyle Polo mentioned the medicinal value of turmeric in the Sino-Indian voyage diary, and in the 13th century, turmeric was introduced from India into European by Arab merchants. During the British rule in India in the 15th century, the curry powder mixed with turmeric and several other herbs were first used for medical purposes in humans. In traditional Chinese medicine, turmeric is a kind of Chinese herbal medicine that promotes blood circulation, releases blood stasis. Turmeric preferably grows in hot and humid climates, and therefore widely existed in tropical and subtropical regions, especially in India, China and Southeast Asia. Modern botany classifies the turmeric, zedoary and yujin into the turmeric family.
Turmeric is mainly composed of curcumin compounds (including curcumin, demethoxycurcumin and bis-demethoxycurcumin), volatile oil and resins [2]. Among them, curcumin separated from the dried rhizomes of turmeric is the main active component of turmeric, with a molecular formula of C21H20O6 (Fig. 1). Curcumin is a tasteless, orange-red photosensitive powder that is insoluble in water. To date, curcumin has shown multiple pharmacological activities, such as anti-inflammation [3], anti-oxidation [4], anti-tumor [5], and immune regulation [6]. Therefore, curcumin is considered a wide range of therapeutic potentials in solid tumors, mental cognitive disorders, and cardiovascular and cerebrovascular diseases. Databases such as Web of Science, PubMed, Google Scholar and CNKI were used to perform the literature search, using terms that define curcumin and disease. The literature search has covered studies including basic research and reviews published in the years from 2000 to 2022. We found that a large number of literature focuses on malignant tumour, Alzheimer's disease, blood abnormalities, and a small part of the curcumin antiviral ability is elaborated. So,with updates of curcumin in anti-tumor, Alzheimer's disease, viral infection and blood disease as the representatives, this paper reviews the pharmacological effects and mechanistic approaches mainly.
In addition to reviewing the outstanding potential of curcumin in a variety of diseases, this paper also focuses on the problems that exist in curcumin from basic research to clinical translation, such as the low solubility of curcumin leads to poor bioavailability, and the difference in the purity of curcumin in the edible and medicinal fields may affect its maximum effect. At the same time, the lack of high-quality clinical research has led to the study of curcumin still staying in the preclinical research era, so the adverse effects of curcumin still need to be explored. Through literature review, we found that there are some case reports, showing that excessive curcumin intake may have adverse effects on the kidneys, heart, liver, blood and immune system, which gives us a warning that curcumin still has a long way to go from basic research to application and transformation.
- In general, the manuscript's novelty and original potential in comparison to the published literature should be described in greater detail in the manuscript. Otherwise, it's a rehash of the curcumin review that was already published.
Reply: Dear reviewer, I'm very sorry, according to your opinion, new references and comments have been added to the article.
- Provide the main mechanism of action/metabolic effects of curcumin in mentioned disorders; you should explain more details on the pathways, enzymes, and markers involved in the articles in the new table for quick review and better comparison of reported studies.
Reply : Dear reviewer, I'm very sorry, according to your opinion, new references and comments have been added to the article.
- Given the wide range of effects reported, it is better to focus on newer studies and of course in clinical studies than pre-clinical studies. Please add the section to the explanation of performed recent clinical trials of Curcumin.
Reply : Dear reviewer, I think your suggestions for adding clinical studies are very important。However, the clinical research progress of curcumin was not very successful, so the part of clinical research was put into chapter 6.3 as one of application hurdles, and the added text was as follows, and the added table (Table1)is shown in the attachment.
The search term "Curcumin" on the website "National Institutes of Health Clinical trial.gov" is available of the 288 clinical studies, 140 clinical studies showed completed, but only 20 had results. Of the 288 clinical studies, 17 were associated with curcumin for cancer and 1 with abnormal blood conditions. Unfortunately, there were no clinical studies of AD or viral infections, and only 3 of these studies had findings. The author draws a table showing the basic information of these 18 clinical studies in the order in which the clinical studies were recruited (Table 1). According to the information in the table, although curcumin has not disappeared from the researcher's sight for 20 years, compared with the preclinical study of curcumin, the clinical study of curcumin is not satisfactory, such as the small sample size, the single type of disease focused (mainly breast cancer, colorectal cancer, prostate cancer, etc.), the difference between curcumin interventions (mainly dose, dosage form, medication time), and the scope of the study endpoints is too large (such as survival, safety, tolerance, etc.). Most importantly, few of these clinical studies have published their findings. This shows that curcumin still has a long way to go to clinical practical application.
- Accurate revision to the text by a native English speaker is required to resolve grammatical and syntax errors and typos.
Reply : Thank you for your suggestion, I have added the cohesion of different content parts in the article, making the topic conversion of the article smoother. I am very sorry, because English is not the author's mother tongue, the English style of this article is rather rough, our team decided to use the English polish service under MDPI to improve the poor English expression.

Reviewer 3 Report
The manuscript in its current form needs critical and substantial improvement. Therefore, I recommend the authors consider the following points during the revision.
- The title of the paper should be adjusted to reflect the content of the paper and its focus. The authors should define in the manuscript the diseases that the paper covers.
- The authors should highlight information about the used databases for collecting and or extracting the data (for example, Web of Science, Scopus, Google Scholar,..) and what keywords were used for the literature search along with the period of covering the collected studies. This is to ensure that the paper covers all available recent and relevant studies. All these points could be highlighted, at least, in the introduction section.
- The introduction section needs more refinement. I recommend the authors add additional pharmacological properties of curcumin such as antiviral properties against herpesvirus infections. This information can be extracted from the reference (DOI:10.3390/microorganisms9020292).
- The reviewed studies and the acquired data should be deeply discussed and interpreted. Unfortunately, the paper in its current form looks like a hasty report, where, the manuscript just reports findings without assessing whether the studies are valid. An attractive review should be based on a critical assessment of the literature published, not just a compilation of the literature sources.
- 5. It would be better to add a new section that discusses strategies or technologies that could be used to improve the reviewed pharmacological properties of curcumin.
- In figure 1, the title should be named ''chemical structure of curcumin'' and not MOLECULAR. The authors should correct this mistake.
- Finally, I recommend the authors double-check the full text for typing and grammatical errors.
Author Response
1.The title of the paper should be adjusted to reflect the content of the paper and its focus. The authors should define in the manuscript the diseases that the paper covers.
Dear Reviewer, what do you think of this title “Benefits and Problems of Curcumin to Human Health”? I think it is a little flat. And the diseases that the paper covers were defined in the new revision version.
2.The authors should highlight information about the used databases for collecting and or extracting the data (for example, Web of Science, Scopus, Google Scholar,..) and what keywords were used for the literature search along with the period of covering the collected studies. This is to ensure that the paper covers all available recent and relevant studies. All these points could be highlighted, at least, in the introduction section.
Reply :Dear reviewer, thank you for your suggestion, the search content has been added in the introduction, as follow.
Databases such as Web of Science, PubMed, Google Scholar and CNKI were used to perform the literature search, using terms that define curcumin and disease. The literature search has covered studies including basic research and reviews published in the years from 2000 to 2022. We found that a large number of literature focuses on malignant tumour, Alzheimer's disease, blood abnormalities, and a small part of the curcumin antiviral ability is elaborated. So,with updates of curcumin in anti-tumor, Alzheimer's disease, viral infection and blood disease as the representatives, this paper reviews the pharmacological effects and mechanistic approaches mainly.
3.The introduction section needs more refinement. I recommend the authors add additional pharmacological properties of curcumin such as antiviral properties against herpesvirus infections. This information can be extracted from the reference (DOI:10.3390/microorganisms9020292).
Reply : Dear reviewer, thank you for your suggestion. Curcumin does have good antiviral activity, but this part was not added to the article considering the length in the previous version. For the sake of the completeness of the review, I added the chapter 5. Curcumin in treatment of virus infection, citing your recommended literature and adding the novel coronavirus related literature as follow.
- Curcumin in treatment of virus infection
Since 2019, the global pandemic caused by the novel coronavirus has been escalating, with more than 6 million deaths currently. Although a wide range of vaccinations have been carried out around the world, the mutation of the virus is very diverse, so the development of drugs to treat the new crown virus is imminent. Recently, Paxlovid and bebtelovimab have been approved for marketing, but clinical evidence is limited and there are skeletal adverse effects, of which bebtelovimab is temporarily listed as an alternative therapy in the NIH guidelines. The potential targets of COVID-19 treatment are mainly spike (S)protein blocking, angiotensin-converting enzyme 2 (ACE2) inhibitor and so on. The novel coronavirus S protein has a very high affinity with the ACE2 receptor on the human extracellular vesicle (EV), and the S protein neutralizing antibody blocks the opportunity for the virus to fuse with the human ACE2 by binding to the viral protein, cutting off the virus host pathway. Wu, et al. extracted EV expressing ACE2 kidney cells and constructed a fake-virus that carries S protein and can bind to ACE2, and found that the EV can block the infection of fake-virus to normal host cells, that is, too much ACE2 can competitively inhibit the binding of viral S protein to human ACE2. The team then used curcumin to stimulate kidney cells that overexpressed ACE2 and found that the EV secreted by the cells could express more ACE2. Transmissible gastro-enteritis virus (TGEV) belongs to the α-coronavirus genus of the coronavirus family and has a certain homology with the novel coronavirus. Li ,et al. treated cells with curcumin in advance and then infected cells with TGEV, and found that curcumin inhibited the proliferation of TGEV and the exression of viral realated proteins in a dose, temperature, and time-dependent manner. Due to the strong contagious of the novel coronavirus, it is difficult to directly use the virus as a research object, but the potential of curcumin to fight the the novel coronavirus has begun to emerge.
Early studies have found that curcumin can inhibit influenza viruses such as human respiratory syncytial virus, H1N1 and H3N3. Curcumin also has a certain inhibitory effect on common human pathogenic infectious viruses. For example, curcumin can significantly inhibit the long terminal repeat (LTR) of the human immunodeficiency virus (HIV) gene under the premise of not affecting cell activity, and affect the virulence and replication ability of the virus. Curcumin can also inhibit Coxsackievirus by reducing viral RNA expression, protein synthesis, degrading ubiquitin-proteasome system (UPS) and reducing virus titer.Wei ,et al. found that after curcumin intervention, the expression levels of hepatitis HBV surface antigen (HBsAg) and hepatitis HBV e antigen (HBeAg) of the HBV stable mutant HepG2.2.15 were significantly down-regulated, and the viral load of intracellular covalently closed circular DNA (ccc DNA) and HBV DNA was decreased simultaneously. Curcumin can also down-regulate HCV gene expression and virus replication by inducing heme oxygenase-1 expression or inhibiting Akt-SREBP-1 pathway. High-risk human papillomaviruses (HPVs) expressing E6 and E7 proteins are important factors for the occurrence and development of cervical cancer and oral cancer Study found that curcumin can down-regulate the nuclear transcription factors AP -1 and NF- κB and selectively inhibit the transcription of HPV-16/E6 proto-oncogene.
In addition to the above common viral infectious diseases, the diseases caused by herpes virus are particularly noteworthy. The herpes virus can form a lifelong infection in the incubation period and has the ability to re-activate periodically. Long-term administration of anti-herpes drugs will lead to various adverse effects and drug resistance. Focusing on natural compounds of plant origin has become a new strategy for the treatment of herpes viruses. It has been found that curcumin exhibits excellent therapeutic potential for anti-herpes virus. Curcumin can affect the recruitment of IE gene promoter by RNA polymerase II mediated by viral transactivator protein vp16 through a pathway that is not dependent on the activity of transcriptional co-activator protein p300/CBP histone acetyltransferase, thereby inhibiting IE gene expression and inhibiting herpes simplex virus -1 (HSV-1) infection. Curcumin may also inhibit HSV-2 replication by down-regulating NF-κB. It has also been found that curcumin can inhibit other herpesviruses as well. For example, curcumin may inhibit human cytomegalovirus by down-regulating IEA, UL83A and Hsp90, may inhibit Kaposi's sarcoma-associated herpesvirus by blocking APE1-mediated redox function, may inhibit the Pseudorabies virus (PRV) transcription of BZLF1 against EBV virus.
4.The reviewed studies and the acquired data should be deeply discussed and interpreted. Unfortunately, the paper in its current form looks like a hasty report, where, the manuscript just reports findings without assessing whether the studies are valid. An attractive review should be based on a critical assessment of the literature published, not just a compilation of the literature sources.
Reply : Dear reviewer, thank you for your suggestions. Clinical studies of curcumin have been added to the new version, pointing out the bottleneck of curcumin clinical translation and making some suggestions as follow
6.3. Lacking of supported clinical reaserch results
The search term "Curcumin" on the website "National Institutes of Health Clinical trial.gov" is available of the 288 clinical studies, 140 clinical studies showed completed, but only 20 had results. Of the 288 clinical studies, 17 were associated with curcumin for cancer and 1 with abnormal blood conditions. Unfortunately, there were no clinical studies of AD or viral infections, and only 3 of these studies had findings. The author draws a table showing the basic information of these 18 clinical studies in the order in which the clinical studies were recruited (Table 1). According to the information in the table, although curcumin has not disappeared from the researcher's sight for 20 years, compared with the preclinical study of curcumin, the clinical study of curcumin is not satisfactory, such as the small sample size, the single type of disease focused (mainly breast cancer, colorectal cancer, prostate cancer, etc.), the difference between curcumin interventions (mainly dose, dosage form, medication time), and the scope of the study endpoints is too large (such as survival, safety, tolerance, etc.). Most importantly, few of these clinical studies have published their findings. This shows that curcumin still has a long way to go to clinical practical application.
5.It would be better to add a new section that discusses strategies or technologies that could be used to improve the reviewed pharmacological properties of curcumin.
Reply : Dear reviewer, thank you for your suggestions. The main reason for the low bioactivity of curcumin is caused by low water solubility, and the original version chapter 6.1.2 provides a rough overview for the current progress of Curcumin complex. Considering your suggestion, I added the length of this part in 6.1.2 as follow.
In this part, we summarized the curcumin embedding complexes with liposomes, polysaccharides or protein, cyclodextrin and other carriers over the past few years and described them as follows, in order to provide a theoretical basis for the optimization of the pharmacological characteristics of curcumin.
Liposomes are emerging nanopharmaceuticals composed of lipid bilayer (phospholipid bilayer and cholesterol) surrounding the internal water environment, which can embed both lipophilic and hydrophilic compounds to improve the solubility and permeability, thus improving the bioavailability. Studies have shown that curcumin liposomes based on phospholipids, cholesterol, Tween-80 can improve the stability of various pH values and metal ions, and also enhance cell absorption and antioxidant properties. Curcumin liposome based on lecithin can prolong the plasma half-life. Curcumin liposomes using phospholipids, sodium hydroxide, ethanol combined can also improve the stability of biological storage. Curcumin liposomes based on soy lecithin, nonionic surfactants, and cholesterol can slow the rate of sustained curcumin release. Borneol-modified curcumin cationic liposomes can significantly increase the contents of curcumin in vivo and in brain tissues and delay the elimination after nasal administration.
In addition to the liposome, curcumin can also be combined with polysaccharides or proteins with biocompatible and low immunogenicity, this kind of biopolymerization nanoparticles can be trapped in the granules by hydrophobic and hydrogen bonding, by inducing cell endocytosis, on the one hand, enhance the direct absorption by the small intestinal epithelial cells, on the one hand inhibit curcumin in the intestine oxidation decomposition reaction, enhance the solubility and bioavailability of curcumin. For example, chitosan-curcumin has better antioxidant properties with metal chelating properties, acetylated starch-curcumin complex has better sustained-release capacity, quercetagetin- curcumin, silk fibroin-curcumin, Pseudostellariae protein- curcumin, rice bran waste- curcumin et.al can improve the water solubility, photosensitivity, and controlled release of curcumin to varying degrees.
Cyclodextrins are starch-derived oligosaccharides with an internal hydrophobic and hydrophilic surface structure, thus enhancing the solubility of hydrophobic compounds in water. Curcumin precursor liposomes can reduce the defect of poor stability and improve the bioavailability in rats by more than 10 times. β-cyclodextrin increases the solubility of curcumin by forming a 1:1 type host-guest complex with curcumin. In vitro experiments have also proved that curcumin encapsulated with cyclodextrin may produce higher blood concentrations of curcumin. Curcumin was also dispersed in a high molecular carrier material to form a microsphere dispersion and a complex with alicyclic acid at 1:1, thus improving the bioavailability and blood concentrations. The curcumin-selenium complex designed have a better therapeutic effect on cadmium-induced liver injury in rats. The drug eutectic formed by the compatibility of Chinese medicine, such as curcumin-artemisinin eutectic and curcumin-ginseng ketone â…¡A eutectic, can also improve the bioavailability of curcumin.
6.In figure 1, the title should be named ''chemical structure of curcumin'' and not MOLECULAR. The authors should correct this mistake.
Reply : Dear reviewer, thank you for your suggestion, the error has been corrected.
7.Finally, I recommend the authors double-check the full text for typing and grammatical errors.
Reply : Dear reviewer, Thank you for your suggestion.

Round 2
Reviewer 1 Report
The authors have satisfactorily addressed the concerns raised in the original version. The revised version is significantly improved. No further concerns.
Author Response
Dear reviewer,
Thank you very much for your careful review and your previous suggestions. I am very grateful.
sincerely,Liu Si-yu
Reviewer 2 Report
Overall, the authors have made significant improvements to this manuscript, and it is now acceptable format. However, before it is taken into consideration for publication in molecules, the following corrections must be made.
1. In the revised manuscript, the author mentioned Table 1. But the manuscript does not contain it. Please check it.
2. A recently published article (https://www.frontiersin.org/articles/10.3389/fphar.2022.820806/full) thoroughly outlined the safety and toxicity profile of curcumin. The citation of this reference by the author will help the readers comprehend the topic under discussion in this manuscript.
3. To give readers more information, the same reference is appropriate to include in section 6.3 as well (https://www.frontiersin.org/articles/10.3389/fphar.2022.820806/full).
4. As a last point, the statement "Further study is warranted" has to be expanded. The conclusions include mostly very general and pharmacologically meaningless statements and instead you need to include a critical assessments of what you have achieved and what are the limitations. What sort of specific future research do you suggest? The underlying issue is this claim that it is a panacea, which could equally be interpreted as a pharmacological irrelevant preparation since it lacks a specific pharmacological activity. So, what are the main challenges in research on curcumin.
Author Response
- In the revised manuscript, the author mentioned Table 1. But the manuscript does not contain it. Please check it.
Reply:Dear reviewer, thank you sincerely for your suggestion. I'm sorry, considering that Table 1 has a large space, it is presented as an attachment.
2.A recently published article (https://www.frontiersin.org/articles/10.3389/fphar.2022.820806/full) thoroughly outlined the safety and toxicity profile of curcumin. The citation of this reference by the author will help the readers comprehend the topic under discussion in this manuscript.
- To give readers more information, the same reference is appropriate to include in section 6.3 as well (https://www.frontiersin.org/articles/10.3389/fphar.2022.820806/full).
Reply:Dear reviewer, thank you for your suggestion sincerely. The author missed these valuable references in literature retrieval , these references have been added to the new version.
4.As a last point, the statement "Further study is warranted" has to be expanded. The conclusions include mostly very general and pharmacologically meaningless statements and instead you need to include a critical assessments of what you have achieved and what are the limitations. What sort of specific future research do you suggest? The underlying issue is this claim that it is a panacea, which could equally be interpreted as a pharmacological irrelevant preparation since it lacks a specific pharmacological activity. So, what are the main challenges in research on curcumin.
Reply:Dear reviewer, thank you for your suggestion. The previous summary was meaningless and insipid and failed to capture the sharp points. The revised version is as follows:
Plant-derived curcumin is still a regular guest to Asian dining tables, and its human health benefits have been developed through pharmacological research over the past three decades. Curcumin seems to be a panacea, due to its multi-target and multi-channel characteristics, it displays a wide range of pharmacological properties in the fields of anti-malignant tumor, AD treatment, blood lipid and blood glucose control, anti-coagulation, anti-platelet, anti-virus and so on. What is shocking is that curcumin can inhibite the novel coronavirus theoretically. However, compared with a large number of exciting pre-clinical studies on curcumin, the clinical studies on curcumin are rather dull. There is no clear clinical study that indicates that curcumin can exactly inhibit the progression of malignant tumors, improve the cognitive function of AD, treat metabolic syndrome, and block virus infection, etc. Curcumin sold on the market exists in the form of health care products or dietary supplements, and aims to relieve alcoholism, protect liver, resist inflammation and relieve pain, in other words, has little to do with the treatment of diseases.
In recent years, focusing the defects of low water solubility and bioavailability of curcumin, more and more curcumin complexes appear, such as being combined with compounds, embeded in special materials and so on, which theoretically enhance the bioavailability of curcumin and even enhance the pharmacological effect. This also clarifies the fact that low purity curcumin is not equivalent to low efficacy. However, few of these curcumin complexes have been truly reflected in the actual application of human health. The safety and toxicity of the compounds and special materials used to enhance bioavailability should be concerned due to the excessive use of additives or colorants harmful to human health.
Although numerous in vivo experiments demonstrated that curcumin at the prescribed dose does not have side effects in the body. However, we cannot take it for granted that curcumin is absolutely safe. The individual case reports of adverse effects manifested in the heart, liver, kidney, blood, reproduction, and immune system cannot be ignored, especially the fact that curcumin may have a cancer-promoting effect on populations with high-risk factors for cancer or precancerous lesions requires high attention. In conclusion, the potential benefits of curcumin for human health are enormous. However, clinical translational application and adverse effects management may pose the greatest challenge for curcumin.
Reviewer 3 Report
Dear authors,
Although the manuscript has been sufficiently improved, the title should be changed as you have suggested "The benefits and problems of curcumin to human health".
Another important point is that you should upload the new revised version of the manuscript without the old references of the old version to avoid any confusion.
Author Response
Dear reviewer, thank you for your suggestion. The title has been revised according to your suggestion. At the same time, to avoid confusion, the old references were deleted.